# Food-Related Carbonyl Stress in Cardiometabolic and Cancer Risk Linked to Unhealthy Modern Diet

**DOI:** 10.3390/nu14051061

**Published:** 2022-03-03

**Authors:** Carla Iacobini, Martina Vitale, Jonida Haxhi, Carlo Pesce, Giuseppe Pugliese, Stefano Menini

**Affiliations:** 1Department of Clinical and Molecular Medicine, “La Sapienza” University, 00189 Rome, Italy; carla.iacobini@uniroma1.it (C.I.); martina.vitale@uniroma1.it (M.V.); jonida.haxhi@uniroma1.it (J.H.); giuseppe.pugliese@uniroma1.it (G.P.); 2Department of Neurosciences, Rehabilitation, Ophthalmology, Genetic and Maternal Infantile, Sciences (DINOGMI), Department of Excellence of MIUR, University of Genoa Medical School, 16132 Genoa, Italy; pesce@unige.it

**Keywords:** advanced glycation end-products, cardiovascular disease, diabetes mellitus, carnosine, food processing, inflammation, obesity, oxidative stress, reactive carbonyl species, tumor disease

## Abstract

Carbonyl stress is a condition characterized by an increase in the steady-state levels of reactive carbonyl species (RCS) that leads to accumulation of their irreversible covalent adducts with biological molecules. RCS are generated by the oxidative cleavage and cellular metabolism of lipids and sugars. In addition to causing damage directly, the RCS adducts, advanced glycation end-products (AGEs) and advanced lipoxidation end-products (ALEs), cause additional harm by eliciting chronic inflammation through receptor-mediated mechanisms. Hyperglycemia- and dyslipidemia-induced carbonyl stress plays a role in diabetic cardiovascular complications and diabetes-related cancer risk. Moreover, the increased dietary exposure to AGEs/ALEs could mediate the impact of the modern, highly processed diet on cardiometabolic and cancer risk. Finally, the transient carbonyl stress resulting from supraphysiological postprandial spikes in blood glucose and lipid levels may play a role in acute proinflammatory and proatherogenic changes occurring after a calorie dense meal. These findings underline the potential importance of carbonyl stress as a mediator of the cardiometabolic and cancer risk linked to today’s unhealthy diet. In this review, current knowledge in this field is discussed along with future research courses to offer new insights and open new avenues for therapeutic interventions to prevent diet-associated cardiometabolic disorders and cancer.

## 1. Introduction

Together with a sedentary behavior, the spread and increase in the consumption of highly processed, calorie-dense foods is leading to an epidemic of obesity and associated cardiometabolic disorders. Relying on high-energy-low-nutrient processed food, the modern maladaptive diet is a major underlying cause of excessive fat accumulation and related chronic inflammation. In turn, adipose tissue inflammation and systemic inflammation lead to several health issues that collectively represent the primary causes of disability and mortality worldwide, such as diabetes mellitus, cardiovascular disease, non-alcoholic fatty liver disease, chronic kidney disease, cancer, and autoimmune and neurodegenerative disorders [1]. Compelling evidence of the role of sterile, low-grade inflammation in chronic diseases comes from randomized trials that have tested drugs targeting specific proinflammatory cytokines, such as tumor necrosis factor-α and interleukin-1β [1,2]. Along the same lines, the Glasgow Inflammation Outcome Study has shown that a combination of markers of the systemic inflammatory response, including high sensitivity C-reactive protein, albumin and neutrophil count could predict 8-year all-cause mortality rate, as well as mortality due to cancer, cardiovascular and cerebrovascular disease [3]. In addition to inducing inflammation through fat accumulation, and adiposopathy [4,5], the unhealthy modern diet (UMD) can trigger pro-inflammatory and proatherogenic changes directly through increased consumption of ultra-processed foods containing high levels of saturated fats, simple sugars, and bioactive carbonyl compounds. The latter are molecules bearing a carbonyl group, i.e., a chemical unit consisting of a carbon and an oxygen atom connected by a double bond. This class of compounds includes reactive carbonyl species (RCS) derived from sugars and lipids, and their reaction products with biological molecules called advanced glycation end-products (AGEs) and advanced lipoxidation end-products (ALEs) [6,7,8,9], respectively.

AGEs/ALEs are a large and heterogeneous group of complex compounds that have both an endogenous and exogenous origin (Figure 1).

Endogenous formed RCS and AGEs/ALEs accumulation induced by hyperglycemia, hyperlipidemia and oxidative stress have been long recognized as critical factors in the pathogenesis of diabetic cardiovascular, renal, and eye complications [10,11]. Most of the harmful effects of these toxic compounds is exerted through the activation of innate immunity receptors such as the receptor for AGEs (RAGE), a 35 Kilodalton transmembrane receptor belonging to the immunoglobulin super family [10,12,13]. In addition, the importance of ingested AGEs/ALEs in insulin resistance and pathogenesis of type 2 diabetes mellitus (T2D) and cancer has been suggested more recently [14,15]. Accordingly, the relevance of dietary preformed AGEs/ALEs as a pathogenic factor in cardiometabolic diseases has been a growing area of investigation [16].

The highly processed foods composing UMD are particularly rich in AGEs/ALEs and their carbonyl precursors, as these toxic compounds, also known as glycotoxins and lipotoxins, are produced through common procedures such as heating, sterilizing, or ionizing [16,17]. However, the role of dietary glyco/lipotoxins in diabetes and diabetes-related vascular complications is debated, as the metabolic fate of most of the ingested AGEs/ALEs and RCS remains unknown. Accordingly, their contribution to systemic carbonyl stress is uncertain and needs further study [9].

Another mechanism linking the UMD and carbonyl stress may concern the macronutrient composition of the diet. In addition to having a high content of RCS and AGEs/ALEs, the UMD is characterized by a high glycemic index, a high glycemic load, and an excessive lipid content [18]. Therefore, it is possible that, regardless of the content of preformed glyco/lipotoxins, UMD can trigger endogenous RCS and AGE/ALE formation by inducing a state of postprandial dysmetabolism, consisting in exaggerated supraphysiological postprandial spikes in blood glucose and lipid levels [19].

Considering all this, the UMD may elicit inflammatory and atherogenic changes directly by favoring systemic carbonyl stress through increased dietary glyco/lipotoxin intake and/or an excess in calorie dense nutrients (i.e., sugars and lipids). After the illustration of carbonyl stress as a pathomechanism contributing to tissue injury in several chronic diseases, we will discuss the evidence in favor of RCS/AGEs/ALEs—both dietary preformed or endogenously formed because of postprandial dysmetabolism—as triggers of systemic carbonyl stress and inflammation. The potential mechanisms linking diet-associated carbonyl stress and cardiometabolic and cancer disorders will also be presented. Finally, we will consider future directions for research to gain new insights on how macronutrient composition of the diet could affect carbonyl stress, and open new avenues for therapeutic interventions aimed at preventing diet-associated cardiometabolic disorders and cancer.

## 2. Biochemistry of Carbonyl Stress and Targeted Therapeutic Strategies

Carbonyl stress is a condition of carbonyl overload due to an excessive production and/or defective detoxification of RCS that leads to an increase in the steady state level of these toxic molecules and, eventually, to the accumulation of their protein adducts and cross-links [11].

### 2.1. Biochemistry

RCS are potent electrophilic molecules carrying a carbonyl group with damaging effects on proteins, nucleic acids, and lipids. Since RCS can be generated by oxidative cleavage and cellular metabolism of both sugars and lipids, the final reaction products of carbonyl stress are called AGEs or ALEs depending on the carbohydrate or lipid nature of the substrate from which the RCS precursor originate [20,21]. However, since AGEs and ALEs present similar structure motifs and activate the same damage mechanisms [12,22], we will use the term AGEs to include all final irreversible products generated by carbonyl stress.

Circulating levels and tissue accumulation of RCS may be the result of increased substrate availability (substrate stress) and/or a generalized increase in the steady-state levels of reactive oxygen species (ROS), that is, oxidative stress (Figure 2).

However, it must be pointed out that carbonyl stress and oxidative stress are not interchangeable terms, and that oxidative stress is dispensable for an increase in carbonyl stress. In fact, RCS overproduction can derive from cellular glycolytic metabolism of excess glucose [23,24,25]. Moreover, if it is true that some RCS may originate from ROS overproduction and the consequent increased oxidative cleavage of sugars and lipids, carbonyl stress may just derive from glyco/lipoxidation reactions driven by increased levels of the oxidizable substrates (i.e., glucose and lipids) in the presence of normal ROS levels [11]. Therefore, substrate stress is a condition sufficient for the increase of the RCS originating not only as normal side products of cellular glucose metabolism, but also as by-products of glycoxidation and lipoxidation reactions, without an increase in oxidative stress (i.e., ROS levels) [11,26]. In addition, chronic overload and deficiencies in the metabolic pathways responsible for the disposal of RCS may contribute to the buildup of these toxic compounds in tissues [11,27].

In metabolically unhealthy subjects, such as diabetic and dyslipidemic patients, RCS overproduction mainly results from an excess of glucose and lipids [23,28,29,30,31], whose circulating levels are chronically increased. For instance, four major RCS and biologically active aldehydes such as glyoxal (GO), malondialdehyde, acrolein, and 4-hydroxynonenal [12,22,32] can be produced by oxidative cleavage of circulating sugars and lipids [29,30]. In addition, in insulin independent cells, such as inflammatory and endothelial cells, excessive glucose flux through the glycolytic pathway leads to formation of toxic glycolytic side products, including the highly reactive α-dicarbonyl compounds 3-deoxyglucosone (3-DG) and methylglyoxal (MGO) [23,24,25,33]. Glycolysis-derived RCS are mainly formed by the spontaneous degradation of the glycolytic intermediates glyceraldehyde-3-phosphate (G3P) and dihydroxyacetone phosphate (DHAP), but also in part from the metabolism of lipids and proteins [34]. Therefore, the rate of RCS formation from cellular metabolic pathways mainly depends on the levels of nutrients (particularly glucose) cells are exposed to [25,33,34,35,36].

By reacting with free thiol groups and amino groups, RCS induce physico-chemical modifications of virtually all biological molecules, thus affecting a number of their functions, including enzymatic activity, ligand binding, half-life, and immunogenicity [37]. In addition to be reliable biomarkers of carbonyl stress and tissue damage [11], the irreversible end-products of these reactions, namely AGEs, can amplify and extend the damage by fueling inflammation and oxidative stress through receptor-mediated mechanisms and the initiation of a chronic inflammatory response [38].

AGEs accumulate in the blood and tissues during aging because of glyco/lipoxidative reactions and the glycolytic metabolism; such buildup is accelerated in several chronic conditions including diabetes, atherosclerosis, obesity, dyslipidemia, renal and liver diseases, and other chronic inflammatory disorders [11,39]. What is more, there is some evidence that dietary glycotoxins may contribute to increase the tissue AGE pool and, as a result, the risk of cardiometabolic disorders [9,40] and cancer. The relationship between AGE intake and cancer has been extensively reviewed in [41,42] and demonstrated in the prospective NIH-AARP Diet and Health Study [43,44].

### 2.2. Targeted Therapies

Overall, as discussed above, carbonyl stress can affect cell and tissue homeostasis through several mediators (i.e., RCS, AGE/RAGE axis, ROS) able to impact on structure and function of various cells and tissues components, as well as cell behavior. In particular, AGE binding to RAGE—or other pattern recognition receptors of the innate immune system [38,45]—activates redox sensitive signaling pathways, including the nuclear factor-κB network, which regulates hundreds of genes involved in cellular stress responses and survival, inflammation, ROS production, fibrosis, apoptosis, proliferation, etc. [46,47,48]. Accordingly, novel carbonyl stress targeted therapies aimed at scavenging RCS (i.e., RCS-sequestering agents) or breaking down the established AGE structures (i.e., AGE breakers) have been under intensive investigation for the prevention of multiple age-related disorders, including cardiometabolic diseases and associated complications [47,48,49,50,51,52,53,54,55,56,57] (Figure 3).

While AGE breakers such as alagebrium primarily act by reacting with and cleaving pre-existing AGEs and protein crosslinks [58,59,60], these anti-AGE compounds cannot prevent structural and functional modifications of the biological molecules by RCS [22,55,61]. In other words, AGE breakers are useful to eliminate the already formed AGEs, thus preventing their pro-inflammatory effect, but cannot prevent the harmful action of newly formed RCS on cell and tissue components. To this end, valid therapeutic strategies are directed at reducing AGE formation by either quenching RCS derived from oxidative and non-oxidative metabolism or enhancing their disposal by induction of detoxifying enzymes. These strategies could reduce both RCS-mediated tissue damage and AGE accumulation. In fact, RCS scavengers and inducers of RCS-detoxifying enzymes have been developed over the past years.

Several compounds with RCS-trapping activity, including vitamin B derivatives (e.g., pyridoxamine), hydrazine derivatives (e.g., aminoguanidine), and amino acid derivatives (e.g., N-acetyl cysteine and L-carnosine), have been tested in preclinical studies with encouraging results, as comprehensively reviewed elsewhere [56,62]. Unfortunately, some of these agents, including aminoguanidine [63] and vitamin B derivatives [64,65,66], have been investigated in clinical trials for vascular complications of diabetes, but have not reaced the clinical stage due to safety concerns and poor efficacy. To avoid safety issues, much more attention has been recently paid to endogenous compounds with proven RCS scavenging activity, such as L-carnosine (beta-alanyl-L-histidine), a naturally occurring dipeptide abundant in the skeletal muscle, nervous system, and kidney [67,68]. Oral supplementation of L-carnosine, or its carnosinase resistant derivatives, has been tested with positive results in several disease models in which carbonyl stress may play a pathogenic role, including diabetes, obesity, and related vascular complications [53,54,55,56,68,69,70,71]. In addition, L-carnosine and its analog carnosinol were also shown to counteract glycolysis-dependent tumor growth by quenching the α-oxaldehyde MGO [72], and preventing the accelerating effect of diabetes on cancer by reducing systemic and tumor carbonyl stress [73].

A possible alternative to target carbonyl stress may be to enhance the body ability to catabolize RCS by using drugs that act on the glyoxalase system (i.e., trans-resveratrol) [74] or the nuclear factor erythroid 2–related factor 2 (NRF2) (i.e., bardoxolone methyl, sulforaphane, etc.), which is a master regulator of cell homeostasis and inducer of phase II detoxification enzymes [75]. Though the mechanisms by which these agents exert their protective action have not been fully elucidated, several in vitro and in vivo studies have shown that glyoxalase and NRF2 inducers are effective against cardiovascular [76,77], neurodegenerative [78,79], and age-related diseases [80,81]. In addition, glyoxalase induction by resveratrol has also shown a favorable effect on glucose and lipid metabolism, inflammation, and insulin resistance [82].

Overall, targeting RCS may provide new therapeutic opportunities to prevent or mitigate diseases in which carbonyl stress is at play. Despite preclinical evidence supports the beneficial effect of glyoxalase and NRF2 inducers in several chronic conditions, there are still few clinical studies, with somewhat inconsistent findings [26]. Conversely, encouraging preliminary results have been obtained in small, randomized clinical trials addressing the effect of the RCS-sequestering agent L-carnosine on obesity and related cardiometabolic disorders [50,51,83,84,85,86]. Another trial on the cardiometabolic effect of L-carnosine is in progress [87]. An advanced preclinical study has also validated the use of the carnosine analogue carnosinol as therapeutic compound, while demonstrating a pathogenic role for RCS in obesity-related metabolic disorders [52]. These studies will be considered later after Section 3 and Section 4 on the diet as possible source of carbonyl stress, and RCS as a potential therapeutic target for diminishing nutrition-related cardiometabolic and cancer risk.

## 3. Dietary AGE Intake, Cardiometabolic Disorders, and Cancer

AGE formation has initially been described as a non-enzymatic reaction between reducing sugars and any amine-containing molecule, including proteins, lipids, or nucleotides. The glycation process is a non-enzymatic multistep chemical reaction that occurs slowly at 37 °C, and it takes several weeks to generate AGEs in the presence of high glucose levels. Therefore, endogenously formed AGEs have always been considered important factors contributing to diabetic complications [88], but not to diabetes development and vascular disease in nondiabetic subjects (Figure 1). This set of chemical processes is also known as Maillard or browning reaction by the name of the French chemist Louis Camille Maillard who, in 1912, discovered them to be responsible for the flavors, colors, and aromas of cooked foods [89]. In fact, exposing foods to high temperature, particularly dry heat (e.g., grilling, baking, frying, or roasting), leads to rapid production of AGEs [90], the amount of which may increase by 10–100 times the levels of uncooked foods [16]. Other thermal intensive food processing procedures, such as industrial sterilization of foods through ultra-high temperature processing, dramatically contribute to increase the food content of AGEs [16,17]. Thus, the amount of ingested AGEs strictly depends on the type of cooking, the degree of industrial processing, but also the type of food (Table 1).

Animal foods that are high in fat and protein, such as red meat, butter, certain cheese, etc., are more susceptible to AGE formation during cooking [16]. In addition, highly processed industrial products, such as bakery products and baked goods, have the highest levels of AGEs, including the GO derivative N(ε)-carboxymethyl-lysine (CML) and the MGO-derivatives N(ε)-carboxyethyl-lysine (CEL) and N(δ)-(5-hydro-5-methyl-4-imidazolon-2-yl)-ornithine (MG-H1) [91]. The role of AGEs in diabetes, cardiovascular disease, and cancer has been proposed in view of the parallel increase of chronic disorders and of dietary AGEs consumption [16,43,44].

Several studies have found that dietary AGEs may have a powerful effect on cardiometabolic health [92,93,94]. What is more, these and other studies showed that circulating levels of AGEs, particularly the GO derivative CML and the MGO-derived AGEs, correlate with indicators of inflammation, oxidative stress, and endothelial dysfunction across all ages [93,95,96,97], even regardless of body weight [93]. Consistently, decreasing circulating AGE levels following a diet low in AGE content was found to reduce markers of oxidative stress, inflammation, and insulin resistance in both healthy subjects and those suffering from metabolic syndrome, diabetes, or kidney disease [98,99,100,101].

Exogenous carbonyl stress, in particular RCS/AGEs derived from the diet, have also been reported to contribute to several type of cancers, including those affecting the breast, pancreas, liver, and kidney [41,42]. Mechanistic studies have shown that AGE structures may alter tumor cell behavior and promote tumor growth by interacting with RAGE and other scavenger receptors on tumor cells [102,103,104,105]. RAGE is expressed in all cell types implicated in tumor formation, including tumor infiltrating inflammatory cells [106], and is linked to the development and progression of several cancers by favoring chronic inflammation [107] and promoting tumor growth and metastasis [108]. Consistently, genetic or pharmacologic blockade of RAGE signaling has been demonstrated to suppress carcinogenesis, cancer progression, and spreading in experimental models [108,109].

According to numerous cross-sectional and intervention studies, circulating AGE levels are influenced by the AGE content of the diet (Figure 4), and a reduction in the consumption of foods with a high content of AGEs, as measured by food records, offers health benefits [100,102,110,111].

However, there are no guidelines regarding safe and optimal intake currently [16], and some evidence even suggests that dietary AGE content may be associated with, but not directly responsible, for the increased circulating levels of these health-threatening compounds [9]. In fact, the real pathological relevance of ingested AGEs remains controversial. In line with the observation that protein modification by AGEs limits their ability to undergo proteolytic digestion [112], a higher fecal excretion of these toxic compounds was observed after consumption of a diet rich in AGEs [113]. Moreover, clear evidence exists that less than 10% of ingested AGEs is found in circulation [114]. The preformed AGEs present in food enter the circulation largely in the free, unbounded form—glycated/carbonylated amino acids and short peptides—to be rapidly excreted with the urine [91]. As a result, AGE intake is positively associated with levels of free plasma and urinary AGEs, but not with their corresponding protein-bound AGEs [91]. The AGE precursors RCS, for their part, due to the high reactivity, are rapidly degraded during the digestion process in the intestine and, therefore, exerts no influence on the levels of RCS and AGEs in vivo [115].

Altogether this implies that dietary RCS and AGEs may not significantly contribute to the endogenous pool of biologically active AGEs, since only protein-bound AGEs can be retained in the body, accumulate in tissues, and induce inflammation and ROS production through receptor mediated mechanisms [9,92,114,116,117]. Collectively, these observations also suggest that the correlation between dietary AGEs and their circulating levels may not be a direct result of AGE absorption. Thus, the known association between UMD, circulating levels of AGEs, and cardiometabolic disorders might be actually mediated by other factors characterizing the UMD, such as the high glycemic index, high glycemic load and excessive lipid content, which may induce supraphysiological postprandial spikes in blood glucose and lipid levels, and transient increase in RCS levels.

## 4. Endogenous, Nutrient-Induced Carbonyl Stress: A New Perspective on Diet in Cardiometabolic and Cancer Risk

It has been established that an unfavorable macronutrient composition of the diet, both quantitative and/or qualitative, can directly elicit acute pro-inflammatory responses and atherogenic changes by inducing excessive glucose and lipid concentration variations and transient disruption of redox homeostasis. Consistently, abnormal increases in blood glucose and lipids after a meal have been associated to transient oxidative stress and an increase in markers of inflammation, endothelial dysfunction, and hypercoagulability in direct proportion to the increases in the circulating levels of glucose and triglycerides [19,118]. On the other hand, inflammation and oxidative stress may promote cancer development and progression [119,120]: it is a shared opinion that diet-related factors, particularly obesity and metabolic inflammation, account for about one third of malignances in developed countries [121].

### 4.1. Endogenous Carbonyl Stress and Cardiometabolic Risk

Epidemiologic evidence suggests that postprandial spikes in blood glucose and lipids are a risk factor for cardiometabolic disorders and may promote atherosclerosis through oxidative stress-dependent endothelial dysfunction, even in nondiabetic individuals [122,123]. Moreover, it has been hypothesized that, if frequently repeated over long periods of time, transient modifications of insulin sensitivity and atherogenic changes characterizing postprandial dysmetabolism [124] can lead to abnormal glucose homeostasis (i.e., impaired glucose metabolism and diabetes) and atherosclerotic cardiovascular disease [122]. Consistently, postprandial glucose fluctuation, calculated as the mean amplitude of glycemic excursion, correlates directly with the ensuing increase in ROS and is an independent predictor of future cardiovascular events even in normoglycemic subjects [125,126]. In addition, postprandial spikes in blood lipid levels enhance the harmful effects of excessive postprandial glucose fluctuations [18,127]. A surge in oxidative stress has been commonly considered the main culprit of the metabolic and vascular effects related with postprandial dysmetabolism [123,126,127], probably because of the inflammatory processes triggered by excessive cellular metabolism of glucose and lipids [126]. However, the actual mediator of the inflammatory response to UMD and the pathophysiological mechanism(s) triggered by postprandial dysmetabolism remain elusive.

Recently, an increased level of glycolysis-derived RCS after a 75-g glucose challenge was demonstrated in individuals with impaired glucose metabolism and T2D [128] (Figure 5).

This finding underlines the possible relevance of carbonyl stress as a candidate to explain the increased risk of vascular complications in diabetic patients with exaggerated postprandial hyperglycemia. What is more, the same study found increased circulating levels of the α-dicarbonyls 3-DG, MGO, and GO after glucose load in subjects with normal glucose metabolism, although to a lesser extent than in prediabetic and diabetic subjects [128]. Importantly, the curve of circulating RCS following the glucose challenge was parallel to that of glucose, and the incremental area under the curve (iAUC) of glucose was strongly associated with the iAUCs of the three α-dicarbonyls. This indicates that, even when the body’s metabolism is still able to cope with the excessive fluctuation of nutrients (e.g., normal glucose metabolism), it may produce a transient excess of RCS. Though beyond the objectives pursued by the study of Maessen et al. [128], these data suggest a potential role of carbonyl stress as a mediator of the association between postprandial dysmetabolism and cardiometabolic risk in metabolically healthy individuals. In fact, as discussed above, in addition to being produced slowly by a multi-step, non-enzymatic reaction between a reducing sugar and proteins, AGEs can be promptly formed endogenously by electrophilic addition reactions that take place between the intermediate metabolites of glucose/lipid metabolism (i.e., MGO, GO and 3-DG) and thiol/amine-containing molecules [129]. Formation of RCS and their derivatives AGEs can be further accelerated by oxidative modifications of excessive circulating sugars and lipids (i.e., glycoxidation and lipoxitation reactions) [15,29,30]. Therefore, substrate stress, although transient, may lead to a buildup of circulating and tissue AGEs, the levels of which tend to accumulate over time because of repeated bouts of nutrient induced carbonyl stress that exceeds the body’s rate of AGE elimination. Overall, these findings suggest that substrate stress-dependent carbonyl stress might be the missing link between postprandial dysmetabolism, sustained chronic inflammatory response and increased oxidative stress, which have been linked to the recent epidemics of prediabetes, diabetes, and cardiovascular disease [16,59,130].

### 4.2. Endogenous Carbonyl Stress and Cancer Risk

Together with their derivatives AGEs, RCS have also been identified as possible contributors to cancer onset and progression, mainly by favoring survival and growth of glycolysis-dependent cancer cells [72,73,131,132]. In particular, the glycolysis-derived α-dicarbonyl MGO is a potent inducer of cell proliferation [133]. A similar effect was reported for other RCS [134], including the lipoperoxidation product 4-hydroxynonenal [135].

A causal relationship between MGO and cancer aggressiveness has been suggested for several types of cancer [136,137,138]. Accumulation of MGO-derived AGEs is a common feature of breast cancer [132,136] and MGO-mediated glycation was shown to promote breast cancer progression and invasiveness by stimulating the remodeling of the extracellular matrix and the activation of the signaling pathways that favor cell survival and migration [136]. Accumulation of RCS derivatives was also found in melanoma cancer [138], and MGO stress was identified as a constant feature of colorectal tumors [137]. By inducing post-translational modification of several lysine and arginine residues and formation of the AGEs CEL and MG-H1, the RCS compound MGO can affect the activity of several regulatory proteins and oncogenic pathways that are known to drive tumor growth and invasion. Among them, the chaperone activity of the heat-shock proteins Hsp90 [72] and Hsp27 [139,140], the kinase activity of the large tumor suppressor 1, and the transcriptional activity of Yes-associated protein [72,73], which is a key downstream target of KRAS signaling [141,142]. Consistently, the activation of multiple oncogenic signaling pathways by MGO was related to the enhanced growth and metastatic potential of several types of cancer in vivo, including breast [72], lung [139], and gastrointestinal [73,137] cancers. Notable is the observation that, in addition to carbonyl stress originated from the enhanced glycolytic activity of tumor cells [72], circulating RCS and related AGE structures (i.e., systemic carbonyl stress) also affect cancer growth, therapeutic resistance, and metastasis, particularly by interacting with RAGE and other receptors on tumor and immune cells [102,103,104]. These findings imply that glucose and lipid spikes after a meal may affect cancer growth through both an increase in tumor carbonyl stress [72,73] and the tumor-promoting effect of systemic RCS and AGEs [15,71] (Figure 6).

In summary, the increased carbonyl stress associated with glucose/lipid spikes may affect cardiometabolic and cancer risk by inducing inflammation, oxidative stress, and endothelial dysfunction, and by promoting cancer cell survival, growth, and migration. Future studies are necessary to analyze the effect of meals with high glycemic index/load and lipid content on endogenous carbonyl stress and establish whether nutrient-induced carbonyl stress may be a mediator of the cardiometabolic, and cancer risk linked to UMD.

## 5. Carbonyl Stress Targeted Therapies to Reduce Cardiometabolic and Cancer Risk

At present, clinical research is moving its first steps to understand the impact of carbonyl stress on metabolic and cardiovascular risk. Diabetes and obesity, particularly abdominal obesity, are carbonyl stress-related conditions [11,143], and weight loss induced by energy restriction and Roux-en-Y gastric bypass were recently found to reduce postprandial dicarbonyl stress [143,144]. A number of small randomized controlled trials to evaluate the cardiometabolic protection provided by L-carnosine have been recently completed [50,51,83,84,85,145], or are in progress [87], with encouraging preliminary results (Table 2).

Most of these intervention studies were designed to assess the effectiveness of the endogenous dipeptide L-carnosine as RCS sequestering agent in vivo, while assessing its efficacy and safety in the treatment of obesity-associated metabolic alterations. The RCS scavenging ability of L-carnosine has been investigated with its oral administration of 2 g/day for 12 weeks. This treatment was effective to improve detoxification of acrolein, one of the most toxic RCS [50]. In this study, although plasma concentrations of both free and conjugated L-carnosine were undetectable, urinary concentration of carnosine-acrolein adducts were positively associated with carnosinuria, a feature that was also observed in non-supplemented subjects [50]. Importantly, a three-month treatment with L-carnosine had positive effects on glucose tolerance and insulin sensitivity in the same population of overweight/obese individuals, suggesting that L-carnosine supplementation may be an effective strategy to prevent T2D [145]. Menon et al. confirmed the improvement in glycemic outcomes by L-carnosine in a recent meta-analysis [147]; the same group later estimated that dietary carnosine supplementation may be a cost-effective treatment option for people with T2D [148]. Treatment with this endogenous dipeptide was also shown to induce changes in plasma lipidome that were associated with increased insulin sensitivity, reduced serum carnosinase-1 activity [84], and favorable modifications in iron homeostasis [83]. Besides improving glycolipid metabolism, L-carnosine supplementation for 12 weeks resulted in reduced oxidative stress and enhanced glycemic control and renal function in pediatric patients with diabetic nephropathy [85].

Overall, these studies suggest a potential role for L-carnosine in the prevention and treatment of obesity-related cardiometabolic disorders and diabetes-induced vascular dysfunction by reducing carbonyl stress and improving glucose and lipid metabolism. Based on these encouraging preliminary data, a randomized, double blind, placebo-controlled trial is in progress to evaluate the effect of L-carnosine intervention on cardiometabolic health and cognitive function in fifty prediabetic and T2D patients [87]. Finally, a recent preclinical study has investigated the efficiency and safety of a novel carnosine derivative, carnosinol (2S)-20(3-amino propanoylamino)-3-(1H-imidazol-5-yl)propanol, which sequesters RCS but cannot be degraded by the circulating carnosinase-1 enzyme that is naturally present in humans [52,149]. Results have shown that carnosinol is highly effective in binding and sequestering both lipid-derived (4-hydroxynonenal and acrolein) and glucose-derived (MGO) aldehydes, while greatly improving metabolism and inflammation in rodent models of diet-induced obesity and metabolic dysfunction [52]. These data extended previous reports showing that L-carnosine ameliorates the manifestation of metabolic syndrome and attenuates the development of T2D in rodent models of these conditions [70,71,146]

Despite the consensus that obesity and a sugar-/lipid-rich diet represent important risk factors for malignancy [121,150,151], to our knowledge there are no studies on the role of endogenous carbonyl stress in human cancer. In addition to the dietary derived preformed AGEs (i.e., exogenous source), whose potential role has been discussed above, the global burden of endogenous carbonyl stress—both tumoral and systemic—includes several RCS derived from oxidative and non-oxidative metabolism of glucose and lipids, such as MGO, GO, acrolein and other aldehydes and ketones [15,133,134,135]. Experimental evidence confirms the causal role of carbonyl stress in cancer, in view of the fact that the RCS trapping agent L-carnosine is effective in reversing the activation of multiple oncogenic signaling pathways and attenuating tumor growth, metastatic potential, and resistance to anticancer treatments [72,136,137,152]. However, most of the experimental studies addressing the beneficial effect of targeting RCS have been carried out on cancer models of rodents fed a normal diet and with a healthy glycolipid metabolism. Thus, the data from these studies can provide information on the role of tumor-associated carbonyl stress due to the increased aerobic glycolysis in cancers [131], but not on the role played by systemic carbonyl stress (i.e., circulating levels of RCS and AGEs). To our knowledge, only two studies have investigated the contribution of systemic carbonyl stress in cancer promotion, showing that targeting RCS with the L-carnosine derivative carnosinol can prevent the accelerating effect of diabetes on pancreatic cancer development and progression [73], and that exogenous AGE administration promotes progression of pancreatic cancer [105]. At present, the role of systemic carbonyl stress associated with postprandial dysmetabolism in cancer remains unexplored.

A separate mention deserves the use of sevelamer carbonate, which was proposed to remove glycotoxins from food, possibly by sequestering AGE-modified proteins in the gut [153]. Sevelamer carbonate is a nonabsorbed phosphate-binding polymer often used in the treatment of end-stage chronic kidney disease, which is known to lower serum levels of calcium and fibroblast growth factor 23 in addition to those of phosphate [154,155,156]. In this setting, sevelamer carbonate was found to reduce coronary artery calcification in hemodialysis patients [157,158]. Besides lowering dietary phosphate uptake, sevelamer was found to improve insulin sensitivity and decreasing LDL cholesterol [159]. Moreover, sevelamer carbonate was effective in improving metabolic and inflammatory abnormalities in patients with T2D and early kidney disease, as attested by the reduced levels of HbA1c, lipids, and markers of inflammation and oxidative stress, irrespective of phosphorus levels [153]. In consideration of its chemical structure and its effect on glycolipid metabolism, it was hypothesized that sevelamer could also bind RCS and AGEs, thus preventing their absorption in the gut [153]. In support of this, a study showed reduced circulating and cellular AGE levels in patients with diabetic kidney disease treated with sevelamer, but not in patients treated with the phosphate binder calcium carbonate [160]. Though the beneficial effect of sevelamer on inflammation and dyslipidemia has been recently confirmed in chronic kidney disease patients [161], there is no convincing evidence that sevelamer is effective in binding AGEs. In fact, sevelamer binds several other compounds in the gut, including endotoxins, bile acids, and gut microbiota-derived metabolites [162]. Altogether, these observations argue against a beneficial effect of this drug through a specific carbonyl stress regulation mechanism.

## 6. Conclusions

The worldwide increase in consumption of highly processed, calorie-dense diet is leading to an epidemic of obesity and cardiometabolic disorders, among which it might be also included cancer [163,164]. A positive energy balance and consequent fat accumulation and dysfunction are well-recognized contributors to this global health phenomenon [165], but food may affect health and disease risk through other ways as well.

RCS and their derivatives AGEs, collectively known as glycotoxins, are neo-formed compounds generated during food processing and preparation. Some studies have suggested that dietary glycotoxins may directly affect metabolic homeostasis, endothelial function, and the immune system [96]. Although present at high levels in several food items characterizing the UMD, the potential biological effect of preformed RCS and AGEs, particularly in healthy subjects, has not yet been definitively demonstrated [9,92] (Figure 7).

The UMD is also characterized by an unfavorable macronutrient composition that can directly elicit acute pro-inflammatory responses and atherogenic changes by inducing exaggerated glucose and lipid excursions and transient disruption of redox homeostasis [19]. The hypothesis of a role of endogenous carbonyl stress as a mediator of the association between postprandial dysmetabolism and cardiometabolic risk is taking ground [128,143]. The postulate is that plasma glucose spikes after a meal rich in readily absorbable carbohydrates, particularly in association with an unfavorable lipid composition [18,127,166], may promote proinflammatory and prooxidant responses by inducing a transient increase in RCS levels and consequent AGE formation. As protein-bound AGEs are not easily eliminated from the body, they can eventually accumulate in vascular and metabolic tissues because of repeated cycles of nutrient-induced-carbonyl stress, thus favoring the establishment of a state of systemic low chronic inflammation.

Consistent with the endogenous carbonyl stress hypothesis, it has been demonstrated that post-challenge glucose excursions are associated with a transient increase in circulating RCS levels, particularly in diabetic and prediabetic individuals [126], and that diet-induced weight loss is associated with decreases in postprandial carbonyl stress in obese subjects [141]. Unfortunately, data on lean and metabolically healthy individuals are limited. Further studies analyzing specific dietary intervention strategies on cardiometabolic and cancer risk markers and large-scale observational studies are needed to gain a better assessment of the risk of nutrient-induced carbonyl stress, and to establish dietary and pharmacological recommendations. Specific areas to be explored include the impact of mixed meals with different glycemic index/load and lipid content on postprandial cardiometabolic status and systemic carbonyl stress, e.g., RCS and AGE formation. It is likewise important to investigate the long-term effect of repeated bouts of nutrient-associated carbonyl stress on AGE accumulation, inflammation, glucose metabolism, and endothelial and vascular function. Such knowledge may help explain at molecular level the protective effects of L-carnosine observed in obese and dysmetabolic individuals. More importantly, this information is needed for optimal design of clinical trials to assess the effectiveness of carbonyl stress targeted therapies in combating the exponential growth of cardiometabolic disorders associated with UMD.

In addition to paving the way for new therapeutic strategies, a better understanding of the pathophysiological mechanisms in cardiometabolic and cancer risk related to UMD should hopefully prompt more effective information campaigns aimed at raising awareness among medical practitioners and the general population about the health risks of the UMD. Only through an integrated approach to nutrition education and the use of new preventive therapeutic strategies can the battle against the epidemic of cardiometabolic and cancer diseases be won.

## Figures and Tables

**Figure 1 nutrients-14-01061-f001:**
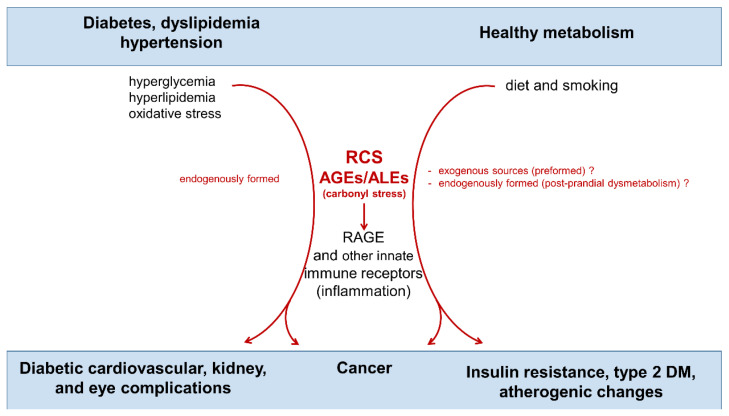
Endogenous and exogenous sources of carbonyl stress and its contribution to cardiometabolic diseases and cancer. Words and arrows in red represent molecules and processes of carbonyl stress. RCS = reactive carbonyl species; AGEs = advanced glycation end-products; ALEs = advanced lipoxidation end-products; RAGE = receptor for AGEs.

**Figure 2 nutrients-14-01061-f002:**
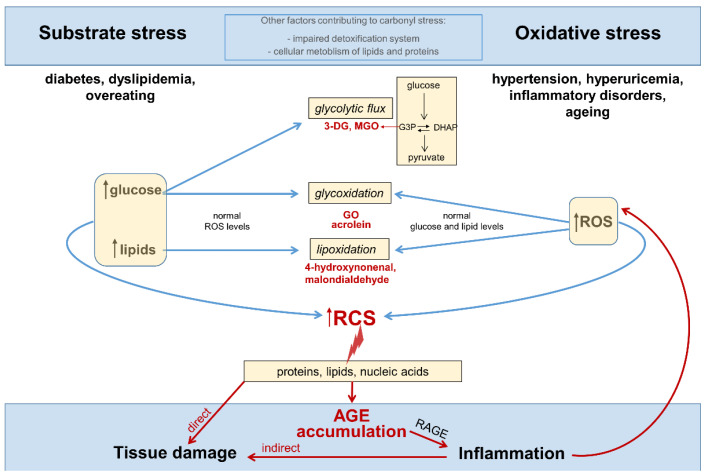
Substrate stress, oxidative stress, and other factors contributing to carbonyl stress, inflammation and tissue damage. Words and arrows in red represent molecules and processes of carbonyl stress. 3-DG = 3-deoxyglucosone; MGO = methylglyoxal; G3P = glycerol-3-phosphate; DHAP = dihydroxyacetone phosphate; GO = glyoxal; ROS = reactive oxygen species; RCS = reactive carbonyl species; AGE = advanced glycation end-products; RAGE = receptor for AGEs; N = nucleus.

**Figure 3 nutrients-14-01061-f003:**
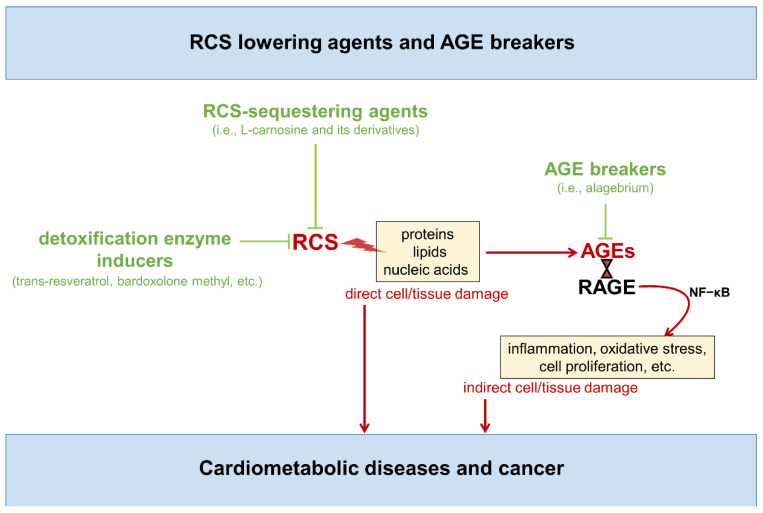
Carbonyl stress targeted therapies Molecules and processes favoring and contrasting carbonyl stress are indicated in red and green, respectively. RCS = reactive carbonyl species; AGEs = advanced glycation end-products; RAGE; Receptor for AGEs; NF–κB = nuclear factor kappa light chain enhancer of activated B cells.

**Figure 4 nutrients-14-01061-f004:**
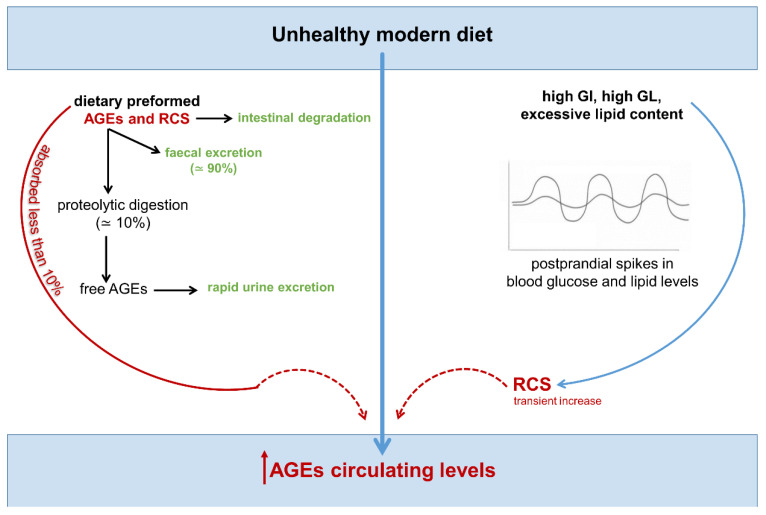
Unhealthy modern diet and increased levels of circulating AGEs: the hitherto unknown relative contribution of dietary intake and post-prandial dysmetabolism. Molecules and processes favoring and contrasting carbonyl stress are indicated in red and green, respectively. Dotted arrows indicate lack of data or definitive evidence. AGEs = advanced glycation end-products; RCS = reactive carbonyl species; GI = glycemic index; GL = glycemic load.

**Figure 5 nutrients-14-01061-f005:**
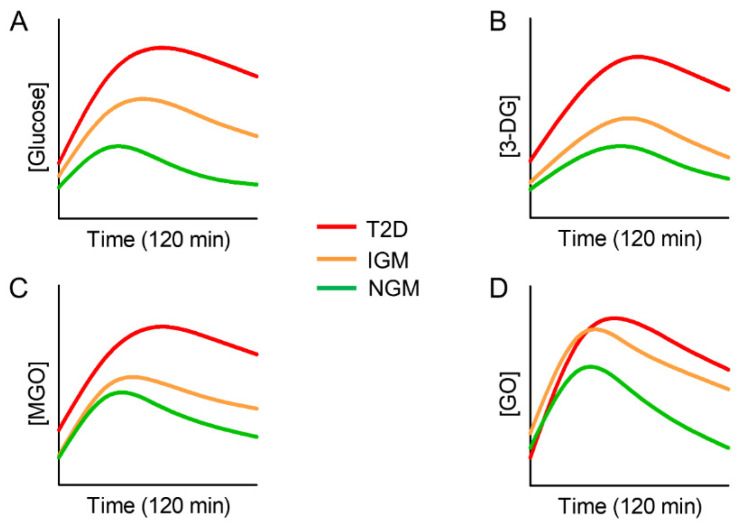
Trend of blood glucose response (**A**) and levels of 3-deoxyglucosone (3-DG, (**B**)), methylglyoxal (MGO, (**C**)), and glyoxal (GO, (**D**)) after a 75-g glucose challenge in individuals with type 2 diabetes (T2D), impaired glucose metabolism (IGM) and normal glucose metabolism (NGM). Adapted from Maessen et al. [126].

**Figure 6 nutrients-14-01061-f006:**
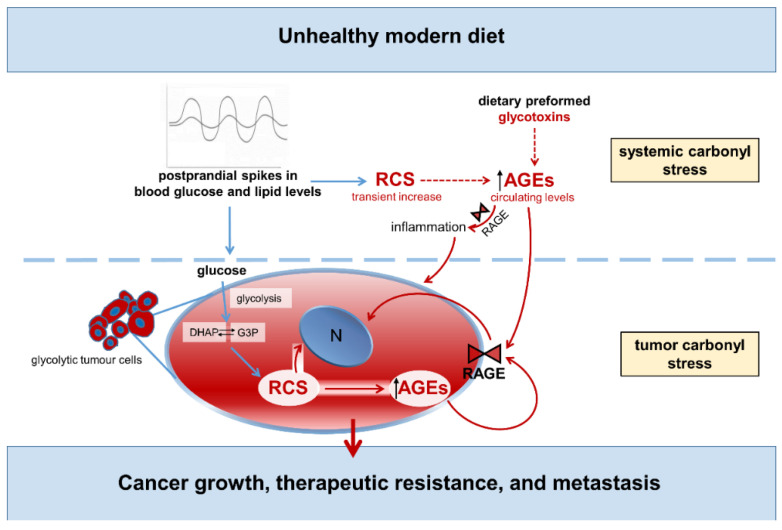
Unhealthy modern diet, carbonyl stress, and cancer. Words and arrows in red represent molecule and processes of carbonyl stress. Dotted arrows indicate lack of data or definitive evidence. G3P = glycerol-3-phosphate; DHAP = dihydroxyacetone phosphate; RCS = reactive carbonyl species; AGEs = advanced glycation end-products; RAGE = receptor for AGEs; N = nucleus.

**Figure 7 nutrients-14-01061-f007:**
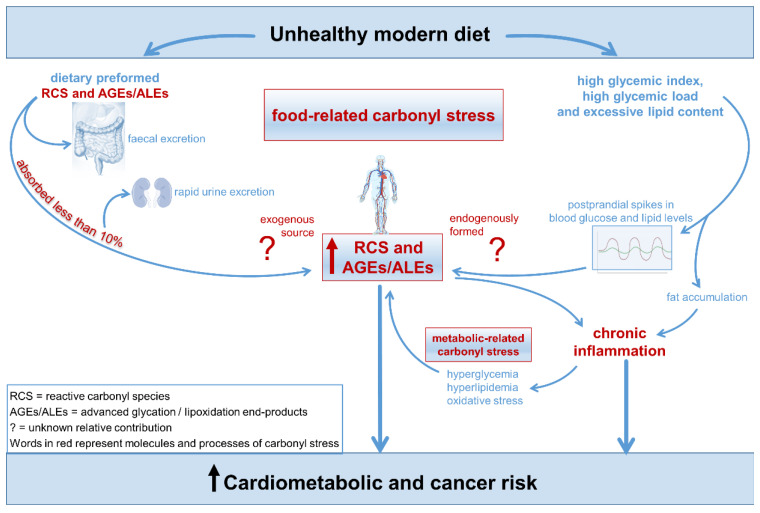
Food-related carbonyl stress in cardiometabolic and cancer risk linked to unhealthy modern diet.

**Table 1 nutrients-14-01061-t001:** The advanced glycation end product (AGE) content in common foods and beverages, based on carboxymethyllysine (CML) content. Adapted from Uribarri et al. [16].

Food Item	AGE kU/100 g	Serving Size (g)	AGE kU/Serving
**Bread/cereals/Breakfast foods/snacks**			
Biscuit	1470	30	441
Bread, white, slice	83	30	25
Bread, white, slice, toasted	107	30	32
Bread, whole wheat, slice	103	30	31
Bread, whole wheat, slice, toasted	137	30	41
Chips, potato	2883	30	865
Corn Flakes	233	30	70
Cracker, wheat	857	30	257
Croissant, butter	1113	30	334
French toast	850	30	255
Muffin, bran	340	30	102
Oatmeal, dry	13	30	4
Pancake, homemade	973	30	292
Pie, apple	637		191
Popcorn	133	30	40
Rice Krispies	2000	30	600
Waffle, toasted	2870	30	861
Popcorn	133	30	40
**Grains/legumes/soy derivatives**			
Beans, red kidney, raw	116	100	116
Beans, red kidney, cooked 1h	298	100	298
Pasta, cooked	112	100	112
Milk, soy	31	250 (mL)	77
Rice, cooked	9	100	9
Soy burger	130	30	39
Tofu, raw	788	90	709
Tofu, broiled	4107	90	3696
**Starchy vegetables**			
Corn, canned	20	100	20
Potato, white, boiled	17	100	17
Potato, white, roasted	218	100	218
Potato, white, french fries	1522	100	1522
**Fruits and vegetables (raw, unless specified otherwise)**			
Apple	13	100	13
Apple, baked	45	100	45
Banana	9	100	9
Carrots, canned	10	100	10
Onion	36	100	36
Tomato	23	100	23
Vegetables, grilled (broccoli, carrots, celery)	226	100	226
**Meat and fish/seafood**			
Beef, raw	707	90	636
Beef, roast	6071	90	5464
Beef, steak, broiled	7479	90	6731
Chicken, breast, raw	769	90	692
Chicken, breast, boiled in water	1210	90	1089
Chicken, breast, roasted	4768	90	4291
Chicken, breast, breaded, fried	9691	90	8965
Lamb, leg, raw	826	90	743
Lamb, leg, broiled	2431	90	2188
Pork, bacon, fried	91,577	13	11,905
Pork, ham, smoked	2349	90	2114
Pork, ribs, roasted	4430	90	3987
Pork, sausage, Italian, raw	1861	90	1675
Pork, liverwurst	633	90	570
Pork, sausage, Italian, BBQ	4839	90	4355
Salmon, raw	528	90	475
Salmon, smoked	572	90	515
Salmon, broiled	4334	90	3901
Shrimp, raw	1003	90	903
Shrimp, fried	4328	90	3895
Tuna, canned with oil	1740	90	1566
Tuna, broiled	5150	90	4635
Turkey, breast, roasted	4669	90	4202
**Milk, milk products, and cheese**			
Cheese, American, white, processed	8677	30	2603
Cheese, brie	5597	30	503
Cheese, cheddar	5523	30	1657
Cheese, feta, Greek,	8423	30	2527
Cheese, mozzarella	1677	30	503
Cheese, parmesan, grated	16,900	15	2535
Cheese, Swiss, processed	4470	30	1341
Milk, whole	5	250 (mL)	12
Pudding, chocolate	17	120	20
Yogurt, vanilla	3	250	8
**Eggs**			
Egg, poached	90	30	27
Egg, scrambled, pan, butter	337	30	101
Egg, omelet, pan, butter	507	30	152
Egg, fried	2749	45	1237
**Beverages**	**AGE** **kU/mL**	**Serving** **size (mL)**	**AGE** **kU/Serving**
Beer	1.20	250	3
Coca Cola	2.80	250	7
Coffee	1.60	250	4
Fruit juice, orange	6	250	14
Tea	1.20	250	3
Wine	11.20	250	28

**Table 2 nutrients-14-01061-t002:** Clinical and preclinical studies on the effects of weight reduction strategies on carbonyl stress, and efficacy of the RCS-sequestering agent L-carnosine (or its carnosinase-resistant derivatives) in the treatment of obesity-associated cardiometabolic abnormalities.

Study	Intervention	Population/Animal Model	Main Outcomes/Purpose
Clinical			
Van den Eynde et al. [143]	calorie restriction	52 abdominally obese men, 25 lean men (18–65 years)	weight loss associated with reduced postprandial iAUC of MGO, GO, and 3-DG in abdominally obese individuals
Maessen et al. [144]	calorie restriction or RYGB	obese women without (*n* = 27) or with (*n* = 27) T2D, 12 lean women	weight loss associated with reduced postprandial α-dicarbonyl levels in diabetic women
Regazzoni et al. [50]	L-carnosine supplementation (2 g/day for 12 weeks)	29 overweight to obese individuals, 8 females and 21 males	increased urinary excretion of carnosine-acrolein adducts (acrolein detoxification),
de Courten et al. [145]	L-carnosine supplementation (2 g/day for 12 weeks), compared to placebo	30 overweight to obese individuals, 15 per treatment arm	reduced fasting insulin and insulin resistance, and normalization of 2-h glucose and insulin after 75-g glucose load
Baye et al. [84]	L-carnosine supplementation (2 g/day for 12 weeks), compared to placebo	24 overweight to obese individuals (13 in L-carnosine, 11 in placebo group)	plasma lipidome changes associated with improved insulin sensitivity and secretion, and low serum carnosinase 1 activity
Baye et al. [83]	L-carnosine supplementation (2 g/day for 12 weeks), compared to placebo	26 overweight to obese individuals (14 in L-carnosine, 11 in placebo group)	iron metabolism changes associated with low serum carnosinase 1 activity and increased urinary carnosine concentration
Elbarbary et al. [85]	L-carnosine supplementation (1 g/day for 12 weeks), compared to placebo	90 patients with diabetic nephropathy	improvement of glycemic control, oxidative stress, and renal function
Baye et al. [87] (Study protocol for an RCT)	L-carnosine supplementation (2 g/day for 12 weeks), compared to placebo	50 participants with pre-diabetes and T2D randomly assigned to the intervention or control group	to analyze changes in metabolic, cardiovascular, and cognitive parameters
Preclinical			
Anderson et al. [52]	Carnosinol supplementation (10 to 45 mg/kg/day for 6 to 12 weeks), compared to placebo	GPx4^+/−^ and WT mice fed a high-fat/high-sucrose diet, and rats fed a 60% high fat diet, compared to chow fed mice and rats	improved glycemic control and muscle insulin sensitivity in mouse models of severe carbonyl stress and diet-induced obesity
Aldini et al. [70]	L-carnosine and D-carnosine supplementation (30 mg/kg/day for 24 weeks), compared to placebo	Zucker obese rat	improved obese-related disorders (dyslipidemia, hypertension, and renal injury)
Albrecht et al. [71]	L-carnosine supplementation (45 mg/kg/day for 18 weeks), compared to placebo	BTBR ob/ob mice (T2D model)	elevated carnosine and carnosine-carbonyl adducts associated with improved glucose metabolism, albuminuria, and glomerular pathology
Al-Sawalha et al. [146]	L-carnosine supplementation (45 mg/kg/day for 16 weeks), compared to placebo	Wistar rats fed a high-fat high-carbohydrate diet (metabolic syndrome model)	reduced blood pressure and glucose, normalized total cholesterol and low-density lipoprotein levels

iAUC = incremental area under the curve; MGO = methylglyoxal; GO = glyoxal; 3-DG = 3-deoxyglucogone; RYGB = Roux-en-Y gastric bypass; T2D = type 2 diabetes; RCT = randomized control trial; GPX4^+/−^ = GPx4-haploinsufficient; WT = wild type; BTBR = black and tan, brachyury.

## Data Availability

Not applicable.

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
