# Peer review of "Food-Related Carbonyl Stress in Cardiometabolic and Cancer Risk Linked to Unhealthy Modern Diet"

_nutrients, 2022, doi:10.3390/nu14051061_

Round 1

Reviewer 1 Report

Overall the structure and contents of the manuscrit fit with the aim. However I would suggest to add a table in the section 3 where the food and dietary intake dose with AGE would be detailed.

Also some comments:

Line 300-309: more references on AGE absorption are needed

Line 344: figure 5 are not extremely needed

Author Response

Reviewer #1

Overall the structure and contents of the manuscrit fit with the aim.

We thank the Reviewer for providing us with positive comments and useful suggestions.

However I would suggest to add a table in the section 3 where the food and dietary intake dose with AGE would be detailed.

As requested by the Reviewer, a Table reporting the AGE content in common foods and beverages has been added in section 3 of the revised manuscript. Both AGE content per 100 g and per serving size have been indicated.

Also some comments:

Line 300-309: more references on AGE absorption are needed

Two new references (#116 and #117) on AGE absorption have been added in the revised manuscript.

 Line 344: figure 5 are not extremely needed

We understand the point of view of the Reviewer 1 that Figure 5 is not entirely necessary, especially in order to convey a quantitative message. However, we think that panels A, B, C, and D may help the reader to follow the argument and understand the text more deeply. Therefore, we have eliminated panel E and left only the graphs in Figure 5.

Reviewer 2 Report

  1. Evidences provided in the article were solid and complete.
  2. the article is well- organized and contained many interesting figures.

Author Response

Reviewer #2

Evidences provided in the article were solid and complete.the article is well- organized and contained many interesting figures.

We thank the Reviewer for her/his overall positive judgement
